# Optimizing an eDNA protocol for estuarine environments: Balancing sensitivity, cost and time

**Thiago M. Sanches**📧*, **Andrea D. Schreier**

Department of Animal Science, University of California-Davis, Davis, California, United States of America

* tmsanches@ucdavis.edu

## Abstract

Environmental DNA (eDNA) analysis has gained traction as a precise and cost-effective method for species and waterways management. To date, publications on eDNA protocol optimization have focused primarily on DNA yield. Therefore, it has not been possible to evaluate the cost and speed of specific components of the eDNA protocol, such as water filtration and DNA extraction method when designing or choosing an eDNA protocol. At the same time, these two parameters are essential for the experimental design of a project. Here we evaluate and rank 27 different eDNA protocols in the context of Chinook salmon (*Oncorhynchus tshawytscha*) eDNA detection in an estuarine environment. We present a comprehensive evaluation of multiple eDNA protocol parameters, balancing time, cost and DNA yield. We collected samples composed of 500 mL estuarine water from Deverton Slough (38°11'16.7"N 121°58'34.5"W) and 500 mL from tank water containing 1.3 juvenile Chinook Salmon per liter. Then, we compared extraction methods, filter types, use of inhibitor removal kit for DNA yield, processing time, and protocol cost. Lastly, we used an MCMC algorithm together with machine learning to understand the DNA yield of each step of the protocol as well as the interactions between those steps. Glass fiber filtration was to be the most resilient to high turbidites, filtering the samples in 2.32 ± 0.08 min instead of 14.16 ± 1.86 min and 6.72 ± 1.99 min for nitrocellulose and paper filter N1, respectively. The filtration DNA yield percentages for paper filter N1, glass fiber, and nitrocellulose were 0.00045 ± 0.00013, 0.00107 ± 0.00013, 0.00172 ± 0.00013. The DNA extraction yield percentage for QIagen, dipstick, NaOH, magnetic beads, and direct dipstick ranged from 0.047 ± 0.0388 to 0.475 ± 0.0357. For estuarine waters, which are challenging for eDNA studies due to high turbidity, variable salinity, and the presence of PCR inhibitors, we found that a protocol combining glass filters, magnetic beads, and an extra step for PCR inhibitor removal, is the method that best balances time, cost, and yield. In addition, we provide a generalized decision tree for determining the optimal eDNA protocol for other studies in aquatic systems. Our findings should be applicable to most aquatic environments and provide a clear guide for determining which eDNA protocol should be used under different study constraints.

**Data Availability Statement:** Data and the analysis pipeline is available on https://github.com/sanchestm/eDNA-Protocol-Optimization.

**Funding:** The authors received no specific funding for this work.

**Competing interests:** The authors have declared that no competing interests exist.

## Introduction

Environmental management relies heavily on knowledge of the spatial distribution of species. In the past decade, environmental DNA (eDNA) monitoring has gained traction as one of the most sensitive and cost-effective monitoring methods [1], allowing researchers to better estimate species occupancy rates in a given habitat. eDNA refers to genetic material present in an environmental sample, which is shed from all organisms in the form of fluids, skin cells/scales, decay and feces from the target species. Filtration or precipitation procedures can isolate eDNA or cells containing DNA from an environmental sample [2]. eDNA acts as a species-specific footprint, which then can be detected by researchers and used to infer the presence of a target species [3]. eDNA might also be used to monitor biodiversity of whole communities as all living beings possess and shed DNA. Although DNA is present in most environmental samples, chemical factors and physical forces can reduce with the DNA detection of an eDNA assay. Unfortunately, due to high variability in the physical and chemical characteristics of studied environments, there are no clear guidelines to assist investigators in choosing an optimal protocol for their particular eDNA monitoring studies.

One environment in which there has been comparably few eDNA studies are estuaries. The estuarine environment provides a challenge for eDNA biomonitoring as the elevated density of solid particles, measured by turbidity levels, can bind to the eDNA and clog the filter pores, limiting the volume of water that might be filtered. Also, estuarine water has been shown to contain elevated levels of PCR inhibitors [4,5]. While we focus on the estuarine habitat here, we assume that if our DNA amplification-based experiments work in these complex conditions, the same approach could also be applied to less turbid freshwater and marine conditions.

The typical protocol used to isolate eDNA from water samples can be described in four steps: filtration, DNA extraction, inhibitor removal and DNA amplification in order to estimate the initial concentration of eDNA [6]. In the filtration step, the water samples, with preferred volumes of 1 liter [7–9], are pressure-pumped through a membrane filter which captures the free DNA as well as tissue and cells suspended in the water. One problem that arises is filter clogging, which leads to under-sampling and irregular sampling volumes. On the other hand, if the pore size or filter material does not capture enough particles, the DNA retention and therefore DNA yield will be reduced. These two diverging issues also have to be balanced with the cost of each filter, which can inflate the total cost of the project considering the number of sampled sites and the number of replicates per site. Currently, the best way to address these problems is to test a variety of filter materials with different pore sizes and identify the ones that have the best characteristics for the sampled system.

The next step is to extract DNA from the filter using conventional extraction methods, which were developed to isolate large nuclear DNA fragments from tissue. However, in the case of eDNA, it is preferable to target small fragments of mitochondrial genes as they have a higher copy number per cell compared to nuclear DNA and are more likely to be detected in an environmental sample. Currently, the standard DNA extraction protocols have been thoroughly optimized for DNA yield, although high cost and low throughput of these techniques are still problematic. Novel extraction protocols aim to address these problems, yet have encountered difficulties in achieving high DNA yield of the standard DNA extraction protocols [10]. In order to increase DNA yield, our study adapted those novel protocols to the context of eDNA such that the sample volumes used are an order of magnitude larger than conventional DNA extractions for molecular biology assays.

Filters may also capture high concentrations of PCR inhibitors. PCR inhibitors are substances that can inhibit PCR amplification. Their inhibiting mechanism varies between

affecting the template DNA, the DNA polymerase or other reagents necessary for the reaction. PCR inhibitors can be catalytic (e.g. proteases degrading proteins and phenol degrading DNA) or work through competitive binding (e.g. melanin forming a complex with the polymerase and humic acid interacting with the DNA template) [11]. Humic matter and proteases are typical PCR inhibitors present in high concentrations in turbid waters and other environmental samples [11,12]. It is often necessary to use a secondary inhibitor removal (SIR) step to further isolate the DNA from contaminants [7]. Since SIR is a column-based extraction, part of the eDNA present in the sample might be lost, as it stays bound to the filter after the last elution step. It is expected that the gain from stopping PCR inhibition outweighs the DNA loss. However, this step might be skipped in order to diminish costs whenever PCR inhibitors are not present in a sample.

Lastly, the isolated DNA is amplified using quantitative PCR (qPCR) with primers specific to the target species, and the initial amount of the target eDNA is determined based on the Cq value [13]. Although it is possible to use conventional PCR, this method significantly underperforms compared to qPCR in terms of sensitivity, and even when DNA is successfully amplified it provides less informative data [14]. Thus, the protocols in this study were only tested using qPCR.

In this study, we separate and optimize four important steps for eDNA biomonitoring of delta estuarine waters, which are characterized by elevated concentrations of solid suspended particles and fluctuating levels of salinity [15]. We targeted Chinook Salmon in our experiments for a variety of reasons. First, as a widespread species in the North American Pacific Northwest, it has invaluable importance for the stability of the marine ecosystem of the region [16] and at the same time, provides a critical source of income for historic fishing communities [17,18]. Little is yet known about the spatial-temporal distribution and estuarine habitat usage of pre-smolt juvenile Chinooks during their annual out migration to the ocean. Developing a high precision, high throughput eDNA protocol optimized for estuarine waters will allow managers to have a better understanding of the habitats used by Chinook in their early life-stages. We comment on the specifics of each step for eDNA biomonitoring, providing a framework that will help investigators make more informed decisions about the best protocol for their study, taking into account various study constraints.

## Methods

### Ethics statement

Holding juvenile Chinook salmon in captivity to sample water for this study was approved by the University of California Davis Institutional Animal Care and Use Committee (USDA registration: 93-R-0433, PHS Animal Assurance A3433-01) under the protocol number #20608.

### Accounting for cost and time in experimental design

To decide on the most practical estuarine eDNA protocol, we first need to determine what it means for a protocol to be efficient. In our case we listed our priorities in the following order: 1) The eDNA yield must be adequately sensitive in realistic scenarios; 2) The protocol must be fast and scalable, and 3) The protocol must be cost effective, considering that reagent cost is the main driver of cost per sample. This order of priorities is influenced by several factors that include species abundance and costs. If the target species is known to be present and potentially at high density, the DNA yield constraint can be loosened, allowing the use of faster and more cost-conscious protocols. If labor cost is inexpensive, choosing a more time intensive yet cheaper protocol will maximize the number of sampling points. On the other hand, in

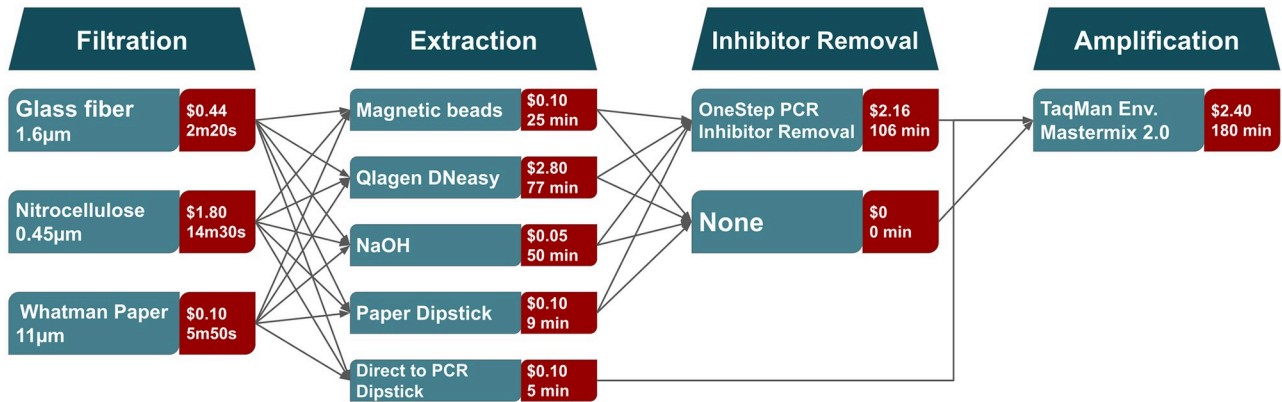

**Fig 1. Scheme of steps for an eDNA protocol with tested methods for each step.** Cost and processing times (in minutes) of each method are shown next to the method name. Number of samples for the measured times varies as the number of samples that can be run in parallel varies between steps. Costs are estimated per sample.

situations where labor accounts for much of the costs, choosing less time intensive protocols will allow more sampling points for the project.

## Experimental design

The goal of this study was to identify the best combination of filter, extraction method, and inhibitor removal steps for studies with varying constraints. We tested three biological replicates for every combination of filter and extraction method, with and without inhibitor removal, and measure the amount of recovered eDNA using qPCR Cq values (Fig 1). A DNA standard curve was developed from a fin clip serial dilution on the same plate. We defined an equation that describes how the efficiency of each step influences the total amount of recovered eDNA:

$$Y \sim Y_f Y_e(f) - I_f I_e * \begin{cases} 0 \text{ , if secondary inhibitor removal is used} \\ \\ 1 \text{ , otherwise} \end{cases} \tag{1}$$

where:

  $Y$: ratio of input eDNA that was amplified by the qPCR
  $Y_f$: ratio of input eDNA that binds to filter
  $Y_e(F)$: ratio of eDNA bound to the filter that is isolated by the extraction method
  $I_f$: filter inhibitor carryover
  $I_e$: extraction method inhibitor carryover
  $I_f I_e$: ratio of input eDNA not available to amplification due to inhibitors

  Then, based on this equation we used Automatic Differentiation Variational Inference (ADVI) [19] to estimate the distribution of the parameters that maximize the likelihood of the observed yields.

## Sampling

To replicate realistic water conditions in terms of salinity, temperature and turbidity while also controlling the presence and amount of Chinook DNA, we combined water samples from a tank containing a high density of juvenile Chinook with an estuarine water sample from a

representative location of pre-smolt Chinook habitat in the San Francisco Estuary. The estuarine water biological replicates consisted of 500 mL of surface water taken with a 1 L measuring cup (sterilized by rinsing in 20% bleach solution and then rinsing in DI water) from Deverton Slough, California (38˚11'16.7"N 121˚58'34.5"W) and collected in a 1 L Nalgene bottle. Next, using another sterile measuring cup, we added 500 mL of tank water known to hold Chinook salmon DNA to each estuarine water biological replicate. The 680 L tank contained 906 Chinook salmon of approximately 11 cm in length. This mixture allowed us to both control Chinook density and observe similar PCR inhibitor levels as those observed in the estuary. In total, we produced 85 samples, which included a deionized water sample negative control, a tank water only positive control and a Deverton Slough water negative control.

## Estimation of average input eDNA

To estimate the average input DNA from the tank water we spiked 10 samples of 1 L surface water from the Deverton Slough, California (38˚11'16.7"N 121˚58'34.5"W) with varying concentrations of isolated Chinook and green sturgeon (*Acipenser medirostris*) DNA totaling 3 samples with 1 ng/L, 3 samples with 0.1ng/L and 3 samples with 0.01 ng/L for the Chinook salmon samples and 3 samples with 10 ng/L, 3 samples with 1 ng/L and 3 samples with 0.1 ng/L for green sturgeon. We tested for green sturgeon concomitantly to validate the protocol for multiple species and verify that probe specificity and detection limit doesn't affect the DNA yield of the protocol. The tenth sample was not spiked and used as a negative control. Then we filtered the samples using a glass filter, extracted the DNA using the Qiagen DNeasy Blood & Tissue Kit (Cat No./ID: 69504) and removed PCR inhibitors using Zymo OneStep™ (Cat No./ID: D6030). Our serial dilution consisted of the same extracted DNA solution used to spike the samples. We estimated the average yield in percentage for this protocol by qPCR amplification. From the Qiagen protocol average yield we could estimate the average input DNA from the tank water. We also estimated that DNA yield percentage is mildly inverse correlated (p-value = 0.0023) to the initial DNA concentration (S1 Fig), while the probability of amplification is logistically correlated to the $\log_{10}$(initial DNA concentration) (S2 Fig).

## Filtration

We filtered the samples one day after sampling to simulate real conditions, where it is not always possible to complete filtration on the same day as sampling. In each filtration run, 4 samples of 1 L were filtered in parallel at the speed of 310 rpm on a peristaltic pump and we timed each filtration event. Filters were folded in half 3 times and stored in a 2 mL microcentrifuge tube and stored at -20˚C. Between runs, tubing and casing were sterilized using a bath of 20% bleach [20], rinsed twice using DI water to remove any remaining bleach and dried.

## Extractions

For all DNA extraction protocols except the dipstick-based ones, we added 180 μL of ATL buffer and 20 μL of 5 U Proteinase K to the microcentrifuge tube and incubated at 56˚C overnight using a rotisserie attachment for 2 mL microcentrifuge tubes. As the incubation time step doesn't require labor, we didn't add it to the total time of the protocol. Next, the filter was compressed inside of the microcentrifuge tube using a pipette tip and the supernatant was transferred to a clean 0.5 mL (NaOH extraction) or 1.5 mL microcentrifuge tube (magnetic beads and Qiagen).

**NaOH-based extraction.**    For each 100 uL of supernatant we added 5.26 μL of 1M NaOH. In a benchtop thermocycler, we incubated the samples at 95˚C for 20 min and ramped down the temperature at a pace of 0.7˚C/min until reaching 4˚C. Next, we added 10% of the total

volume of 1M Tris-HCL. Samples were vortexed and centrifuged for 15 min at 4680 rpm. Without disturbing the pellet, 100 μL of the supernatant was extracted and transferred to a new 1.5 mL microcentrifuge tube and stored at -20˚C.

**Magnetic beads.** For each sample, 180 μL of Agencout AMPure XP (Beckman Coulter™; Cat No./ID: A63881) was added to the solution and incubated at room temperature for five minutes. Then the microcentrifuge tubes were placed onto the magnetic plate (DynaMag™-2; Cat No./ID: 123.21D) for 2 minutes. We removed the supernatant and washed the magnetic beads twice using 200 μL of a freshly made 70% ethanol solution with an incubation time of 30 s in the magnetic plate. We then air-dried the beads for 3 minutes. A total of 100 μL of TE solution was used to resuspend the particles and elute the DNA. The solution was incubated for 1 minute at room temperature before pulling down the magnetic beads with the plate for 2 minutes. Lastly, the supernatant was transferred to clean 1.5 mL microcentrifuge tubes.

**Qiagen DNeasy cell and tissue.** The Qiagen DNeasy extraction was performed following the manufacturer's recommendations. A total of 200 μL of AL buffer was added and the samples were incubated at 56˚C for 10 minutes. We added 200 μL of ethanol to each sample and the solution was transferred to the column and centrifuged at 8000 rpm for 3 minutes. Then the column was washed using 500 μL of Wash Solution N˚1 and centrifuged at 6000 rpm for 1 minute. Then the column was washed again with Wash Solution N˚2 and centrifuged for 3 minutes at 1400 rpm. Next, 100 μL of AE solution was added and incubated for 20 minutes before centrifuging at 8000 rpm for one minute. The flowthrough was then stored at -20˚C

**Whatman paper dipstick.** The Dipstick method was tested due to its cost efficiency, short processing time, and the fact that it allows for eDNA extraction in the field at the expense of DNA yield. Dipsticks were made following the protocol described in [10]. We used the qualitative Whatman filter n˚1 to make our dipsticks and used an effective surface area of 8 mm$^2$ (2 mm width and 4 mm height). We added 200 μL of lysis buffer and ground the filter using a pipette tip until the filter was dissolved. Then we dipped the dipstick in the lysis buffer solution (20 mM Tris [pH 8.0], 25 mM NaCl, 2.5 mM EDTA, 0.05% SDS) 3 times, then dipped 3 times in 100 μL of wash solution (10mM Tris [pH 8.0], 0.1% Tween-20), and 3 times in a final solution of nuclease free water which then was stored at -20˚C. In the case of "straight to qPCR" dipstick extraction, we directly dipped the dipstick after the wash step into the qPCR reaction. An advantage of the "straight to qPCR" dipstick is that, by skipping an elution step, we avoid dilution of the DNA bound to the dipstick. DNA captured on the dipstick is at a higher concentration than what would be eluted in nuclease free water. Though this method avoids a dilution step, PCR inhibitors will also be at a higher concentration and could lead to variable results from multiple qPCR reactions from the same sample.

## Secondary inhibitor removal

Zymo OneStep™ PCR Inhibitor Removal Kit (Zymo Research; Cat No./ID: D6030) was used following the manufacturer's protocol in order to remove any carryover PCR inhibitors from previous steps. We added 600 μL of Prep-solution to the column and centrifuged at 8000 g for 3 minutes, the flow-through was discarded, then 50 μL of DNA elute from previous steps were added to the column and centrifuged at 16000 g. The flowthrough was then stored at -20˚C.

## qPCR amplification oligos

Quantitative PCR detection for Chinook salmon was developed by adapting the protocol from [21]. Reaction solution totaling 20 μL was composed of 1× TaqMan™ Environmental Master Mix 2.0 (ThermoFisher Scientific; Cat No./ID: 4396838), 0.9 μM concentration of each primer, and 0.7 μM of the Taqman probe, and 6 μL isolated DNA extract from previous steps. The

**Table 1. List of DNA oligonucleotides used in this study.**

| Oligonucleotide | Sequence |
|---|---|
| Probe sequence | FAM-5′-AGCACCCTCTAACATTTCAG-3′-ZEN / Iowa Black FQ |
| Forward primer | 5′-CCTAAAAATCGCTAATGACGCACTA-3′ |
| Reverse primer | 5′-GGAGTGAGCCAAAGTTTCATCAG-3′ |
| Gblock sequence | 5′-ACCATCGTTGTTATTCAACTACAAGAACCT AATGGCCAACCTCCGAAAAACCCATCCTCT CCTAAAAATCGCTAATGACGCACTAGTCGA CCTCCCAGCACCCTCTAACATTTCAGTCTG ATGAAACTTTGGCTCACTCCTAGGCCTATG TTTAGCCACCCAAATTCTTACCGGGCTCTT CTTAGCCATACACTATACCT-3′ |

Primers were used for DNA amplification. Probe was used for the qPCR step for DNA quantification. Gblock was used for creating a standard ladder for the qPCR reaction and made possible the conversion from initial DNA concentration to copy number.

chosen primers, probes and gblock were designed following [21] and are shown in Table 1. Thermocycling was performed on a Bio-Rad CFX96 real-time detector using the following profile: 10 min at 95˚C, 40 cycles of 15s denaturation at 95˚C and 1 min annealing–extension at 60˚C.

## Data analysis

Data analysis was performed in Python 3.7 and the analysis pipeline is available on https://github.com/sanchestm/eDNA-Protocol-Optimization. We measured interference between filter type and extraction method using two competing models, one that includes the interference effect and one that does not. Using ADVI inference we fitted the data to the models [19]. From the ADVI fitting for the best model we estimated the distribution of filter eDNA yield percentage, extraction eDNA yield percentage and PCR inhibitor carryover for filtration and extraction. To estimate which step of an eDNA experiment has the most variance between methods, and therefore can lead to the most significant gains when optimized, we trained a random forest regressor [22] with the collected data and estimated importance of each step of the experiment.

## Results

### Effect of filter type on DNA extraction yield

We first examined different methods for the initial two steps of an eDNA protocol, filtration and DNA extraction, and tested for interference between these steps. In general, when optimizing a protocol consisting of several steps, it is important to identify if previous steps interfere with the effectiveness of subsequent steps. In our study, the main possible interference is between the filter used and the extraction method. The yield percentage of a certain extraction method could change depending on which filter was used. Possible reasons for interference between filter and extraction method include different particles binding differentially to filters and extraction methods not isolating DNA from all types of particles at the same yield percentage. The models that we tested are the following:

Model with interference:

$$Y \sim Y_f(Y_e - interference(F)) - I_f I_e \begin{cases} 0 \text{ , if secondary inhibitor removal is used} \\ 1 \text{ , otherwise} \end{cases} \quad (2)$$

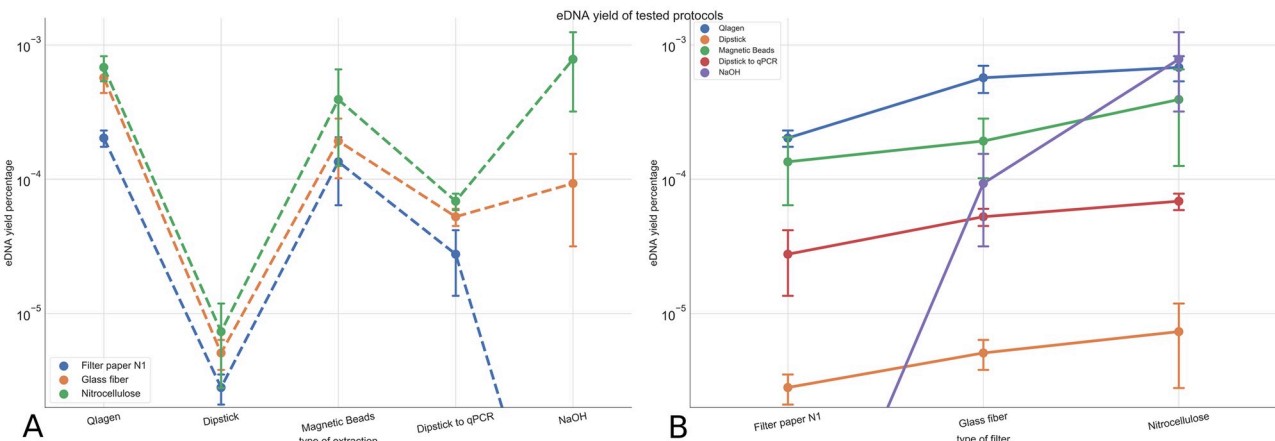

**Fig 2. Relationship between total DNA yield and filtration and extraction protocols.** (A) No crossing between lines indicates that on average, DNA yield ranking for the filters is independent of the extraction protocol. Error bars represent the 95% confidence intervals. The combination of Whatman filter and NaOH extraction wasn't able to amplify the target Chinook DNA. (B) No crossing between lines indicates that on average, DNA yield ranking for the extraction method is independent of the filter type. The NaOH extraction protocol is the only case where the ranking order is not maintained and can be explained by the added effects of carry-on inhibitors.

Model without interference:

$$Y \sim Y_f Y_e - I_f I_e * \begin{cases} 0 \text{ , if secondary inhibitor removal is used} \\ 1 \text{ , otherwise} \end{cases} \quad (3)$$

For both widely-applicable information criterion (WAIC) and Leave-one-out cross validation (LOO) the model without interference was selected with a weight of 1 in both cases [10,23]. Other evidence for the absence of interactions is that the ranking order of filter yield (1st *Cellulose nitrate - 2nd Glass fiber - 3rd Filter paper N1)* does not change independently of which extraction method is chosen (Fig 2A). Similarly, the yield ranking for extraction methods is not affected by filter choice. (Fig 2B). Only the NaOH method breaks the independence rule for the nitrocellulose filter. In this case, target DNA could not be amplified from NaOH extractions without secondary inhibitor removal, resulting in an upwards skewed average of the DNA yield as the samples without secondary inhibitor removal were not taken into account.

## DNA retention and filtration time per filter type

Next, we compared DNA yields from three different filters. The nitrocellulose filter outperformed the glass fiber filter in terms of DNA yield by 1.6 times and the Whatman n˚1 filter by 3.75 times on average (Fig 3). The percentage of captured eDNA copies were 0.00045 ± 0.00013, 0.00107 ± 0.00013, 0.00172 ± 0.00013 for paper filter N1, glass fiber, and nitrocellulose respectively. In other words, 1.6 L and 3.75 L of water would need to be filtered through a glass fiber filter or Whatman n˚1 filter, respectively, to isolate the same amount of DNA as filtering 1 L of water through a nitrocellulose filter. However, the glass filter outperforms the nitrocellulose and Whatman filters in terms of filtration time, with the glass filter not only being drastically faster but also more consistent and resilient to variations in turbidity (Fig 4). Filtration times were 2.32 ± 0.08 min, 14.16 ± 1.86 min and 6.72 ± 1.99 min for glass fiber, nitrocellulose and paper filter N1, respectively.

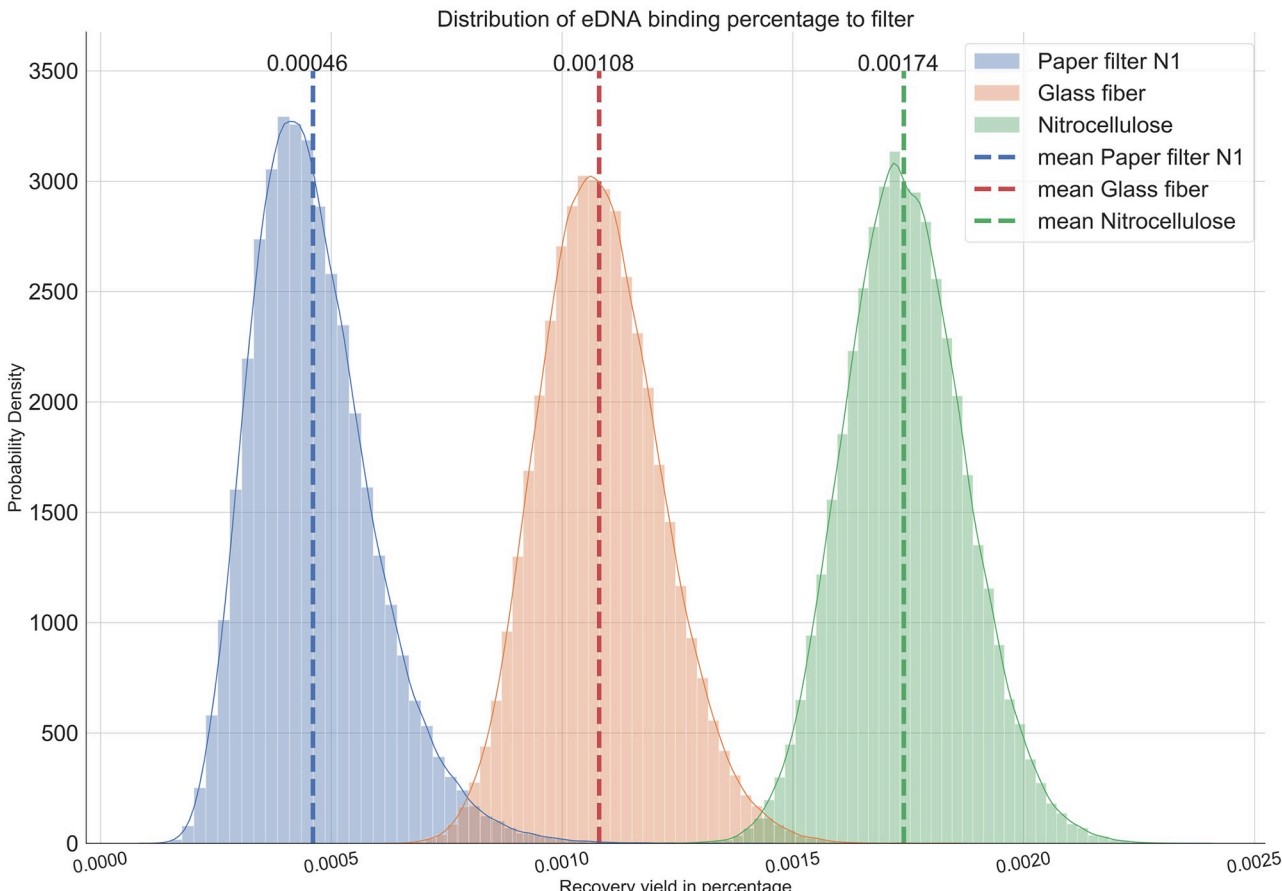

**Fig 3. Distribution of DNA capture ratio for each filter type from the Automatic Differentiation Variational Inference model.** The broadness of the curve shows the variability of the ratio of the input DNA that binds to the filter. The peak of each distribution is the mean yield ratio of DNA recovery for that filter type. The nitrocellulose filter yielded the highest recovery ratio with little efficiency overlap compared to glass and Whatman filters.

## Comparison between DNA extraction protocols

All extraction methods could yield enough eDNA to be detectable by qPCR amplification. The Qiagen DNEasy kit had the highest DNA yield, outperforming NaOH by 1.7 times, magnetic beads by 2.26 times, direct to qPCR dipsticks by 9.71 times and regular dipsticks by 358 times (Fig 5). Observed DNA extraction yields in percentage were 0.475 ± 0.036, 0.047 ± 0.037, 0.287 ± 0.037, 0.206 ± 0.037, 0.132 ± 0.053 for QIagen, dipstick, NaOH, magnetic beads, direct dipstick, respectively. At the same time, the Qiagen kit is by a considerable margin the most time-consuming method, requiring 77 minutes to process 18 samples. In contrast, the direct to qPCR dipstick approach was the fastest and most cost-efficient method by a wide margin.

## Importance of each protocol step on total DNA yield

The extraction method was shown to be the most influential factor for the eDNA yield from the random forest aggressor analysis (Fig 6). Therefore, further optimization experiments should focus on this step, experimenting with different protocols to extract the eDNA in order to maximize protocol eDNA yield. Meanwhile, in the context of our experiments, the removal

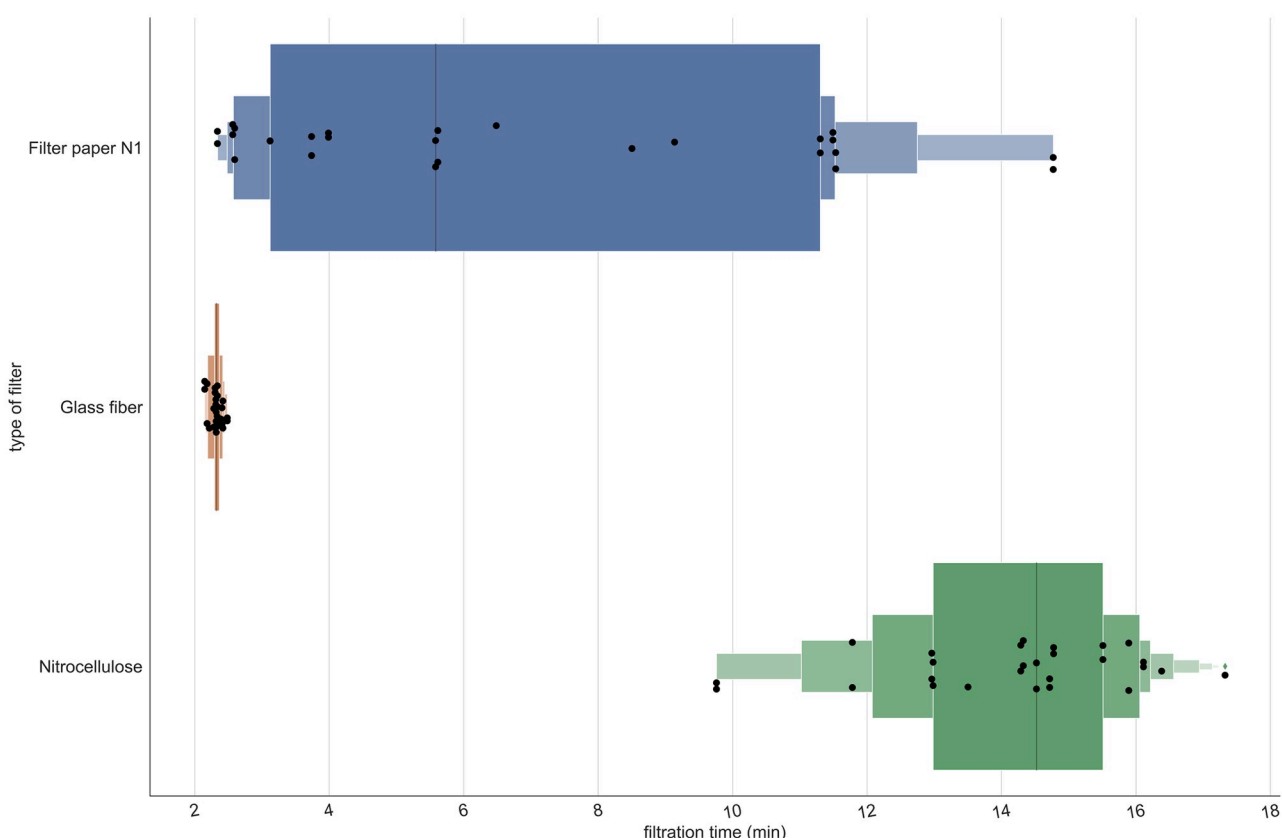

**Fig 4. Percentiles and median filtration time in order to filter 1 L of estuarine water for each filtration method.** The glass filter outperforms nitrocellulose and Whatman by a significant margin in terms of average filtering time and consistency in the filtering time. Dots are filtration events while the black line represents the median value filtering time. Boxes indicate 10% quantiles.

of inhibitors was shown to have little impact to the total DNA yield estimated by qPCR, although published data [7] have shown that inhibitor removal highly influences the amplification probability of the qPCR reaction.

### Conditions in which the use of SIR is necessary

We observed that the secondary inhibitor removal step always outperformed skipping this step. Regressions from Fig 7A and 7C were always positive and the distributions from Fig 7B and 7D were always greater than zero. The nitrocellulose filter and the NaOH extraction were the methods that carried the most PCR inhibitors, while the other methods for each step showed a high overlap of their carryover inhibitor distributions. Secondary inhibitor removal was essential to observe any amplification using the NaOH extraction method, which also suggested that this method is inefficient at removing PCR inhibitors.

### Discussion

As the eDNA field has progressed and we've learned more about how to detect DNA under different environmental conditions, the number of methods for each step of an eDNA assay test has exponentially increased. Due to the wide array of options, identifying the best protocol given the environment and the target species of the survey is complex. Several publications

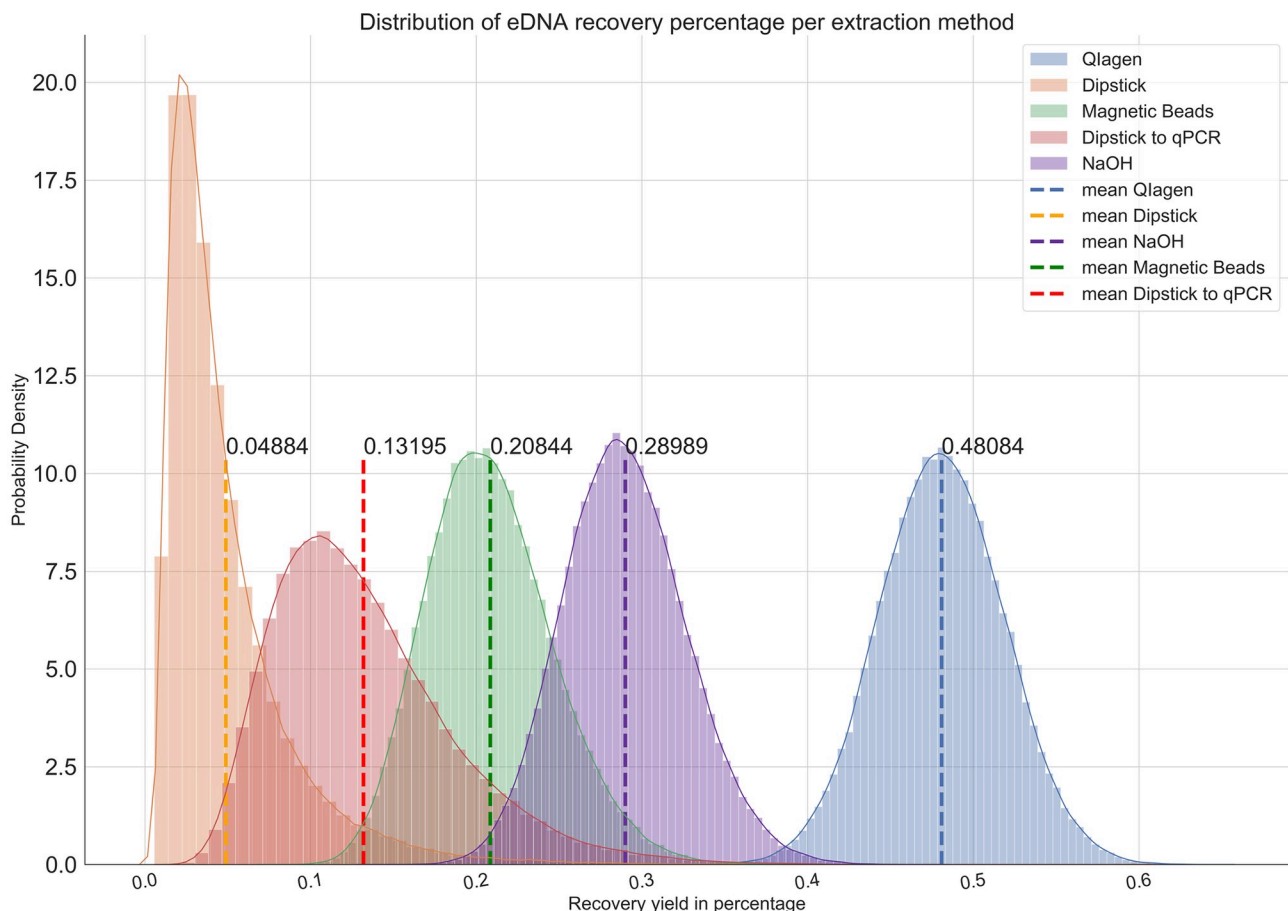

**Fig 5. Modelled distribution of percentage yield for each extraction protocol.** The width of each of the curves shows the variability in modelled yield. The peak of the distribution is the mean yield per extraction type. Qiagen DNeasy yields the best yield with little overlap with other methods. Meanwhile magnetic beads and NaOH have shown similar distributions with significant overlap, while both dipstick methods underperform the other methods. It is important to note that this plot does not take into account PCR inhibitor carryover, which might vary significantly between methods.

describe DNA yield comparisons between different methods [10,24–26]. We chose to test nitrocellulose, glass fiber and paper filter N1, as in the literature nitrocellulose consistently presents the highest DNA yield [27,28], while glass fiber is the most common filter material [2,29] and paper filter had shown to have high DNA binding [10] while it's cost is a fraction of other materials. The chosen pore size for glass fiber was chosen based on [27], while nitrocellulose pore size was chosen relative to the most common pore size [30], while paper filter pore size was chosen to maximize flow. We opted to test more extraction methods than filter material as we expected that most filter materials capture particles indiscriminately in a similar fashion, while extraction methods do vary significantly on the mechanics of the protocol.

One of the novel observations from our experiment is that each step of an eDNA assay seems to work independently of the previous step. The lack of interference between steps shows that for future optimization tests, it may not be necessary to test all the possible combinations of filters and extraction methods at the same time. Instead, one might test each section of the protocol independently and still obtain an optimal eDNA assay protocol. This would allow more methods to be tested for each step and increase the number of replicates for each method in future optimization experiments. The observed low levels of interaction between

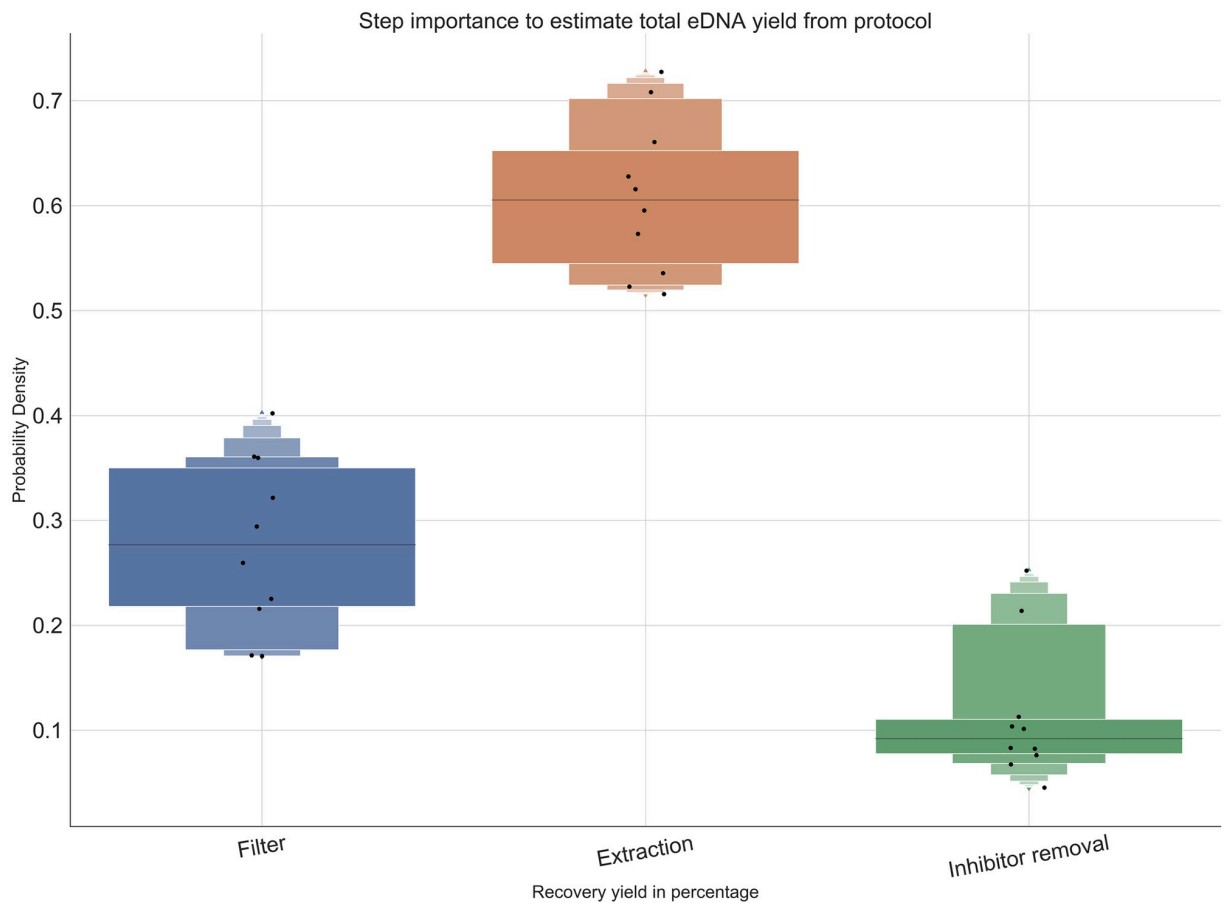

**Fig 6. Influence of each eDNA protocol segment to total eDNA yield ratio estimate.** The extraction method was the factor that had the highest influence on the total eDNA yield of the total protocol, while inhibitors didn't have a significant impact compared to the other components of the total protocol.

steps also permits a better comparison to the literature. As an example, both this paper and [27] test approximately 1 μm glass fiber filtration. In [27], the 1 μm filter retained the most DNA, while the nitrocellulose filter had a better DNA retention than the 1.6 μm glass fiber filter in our experiment. Even though other steps of the assay might differ between the publications, we can still conclude that nitrocellulose outperforms glass fiber in any mesh size tested in [27].

Estuaries possess unique physical characteristics and biodiversity of major importance for both marine and riverine ecosystems. The Chinook salmon is a perfect example of a species in which the understanding of its annual out migration patterns in these estuaries are essential for their preservation. In order to understand the migration patterns of Chinook Salmon, we optimized an eDNA protocol for estuaries. For estuarine conditions, we concluded that glass fiber filters are the most efficient because estuaries possess high turbidity, which clogs the other tested filters. The high filtration time for the Whatman and nitrocellulose filters hinders the throughput of the sampling, possibly leading to under sampling in the total eDNA survey. For the DNA extraction, we considered magnetic beads to be the optimal method for estuarine waters, as dipstick methods had subpar DNA yield while Qiagen DNeasy did not allow high throughput sampling due to cost and processing time. Even though Ampure XP has an elevated cost per sample, we are able to reduce this cost 100-fold by making a magnetic beads

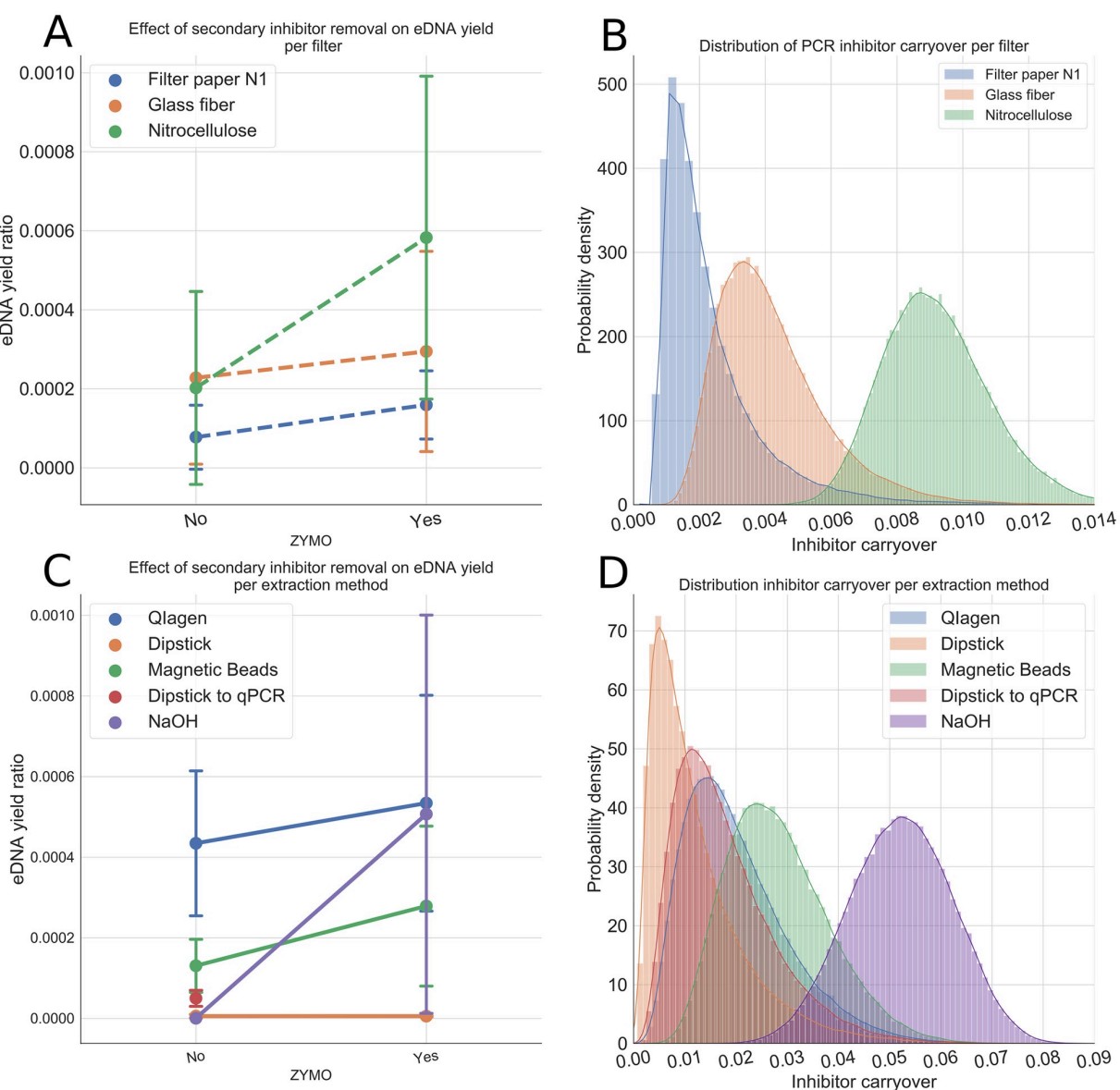

**Fig 7. Effects of adding a secondary inhibitor removal step to the eDNA estimation protocol.** (A-B) DNA yield variation from using a OneStep PCR Inhibitor Removal©. The nitrocellulose filter produced the highest inhibitor carryover levels at the same time it captured the highest percentage of free eDNA. This suggested that the nitrocellulose filter captured particulates indiscriminately with high efficiency (C-D) Estimated distributions for inhibitor carryover for filter and extraction method. Aside from NaOH extraction, other methods had similar distributions of carryover PCR inhibitors with high overlap. Therefore, NaOH extraction, even if it has an elevated eDNA yield, doesn't properly address the high levels of PCR inhibitors commonly encountered in environmental samples.

solution in-house [31]. Buying in bulk is also another alternative to reduce costs, though that might be limited to the initial funding of the project.

Our experiments have shown that the use of secondary inhibitor removal has little influence on total eDNA yield. This observation may be explained by several nonexclusive factors. First, inhibitors generally work by binding to DNA strands and do not act as catalysts. Therefore, if the ratio between eDNA:inhibitors is significantly elevated, which is expected in tank experiments, we would predict minimal effects of the inhibitors. Another possibility is that during the time we

were sampling, there were fewer inhibitors than usually observed in estuaries. Furthermore, fil-
tration and DNA extraction methods vary considerably in DNA yield, more than the observed
effect from SIR. Lastly, PCR inhibitors might not affect the eDNA retrieval but only the probabil-
ity of amplification. This last observation might also explain why the probability of amplification
and DNA yield is not always fully correlated. Therefore, considering this experiment's results
and previous findings [32,33], we advise the use of secondary inhibitor removal, if possible, as it
improves the DNA yield and amplification probability in the context of estuarine samples.

Our study suggests that in most cases, using a glass fiber filter and magnetic beads would be
the most practical method to generate the maximum amount of information obtained about
fish distribution. We also concluded that DNA extraction from the filters is the most time-con-
suming step and most variable in terms of efficiency. Therefore, this is the step that should be
decided with utmost care in order to maintain the high-throughput and useful detection limit
of the desired methodology. For this reason, magnetic beads DNA extraction is a promising
alternative to silica column extraction, as this method strikes the balance between yield, ampli-
fication probability, carryover PCR inhibitors and time to process samples. Meanwhile, the
cost of using magnetic beads can be mitigated by developing the necessary reagents in-house.

However, different protocols may yield better results under certain circumstances, such as
when the target species is extremely rare or ubiquitous. In order to choose the best protocol,
we constructed a simple decision tree for those scenarios (Fig 8). We also ranked the protocols,

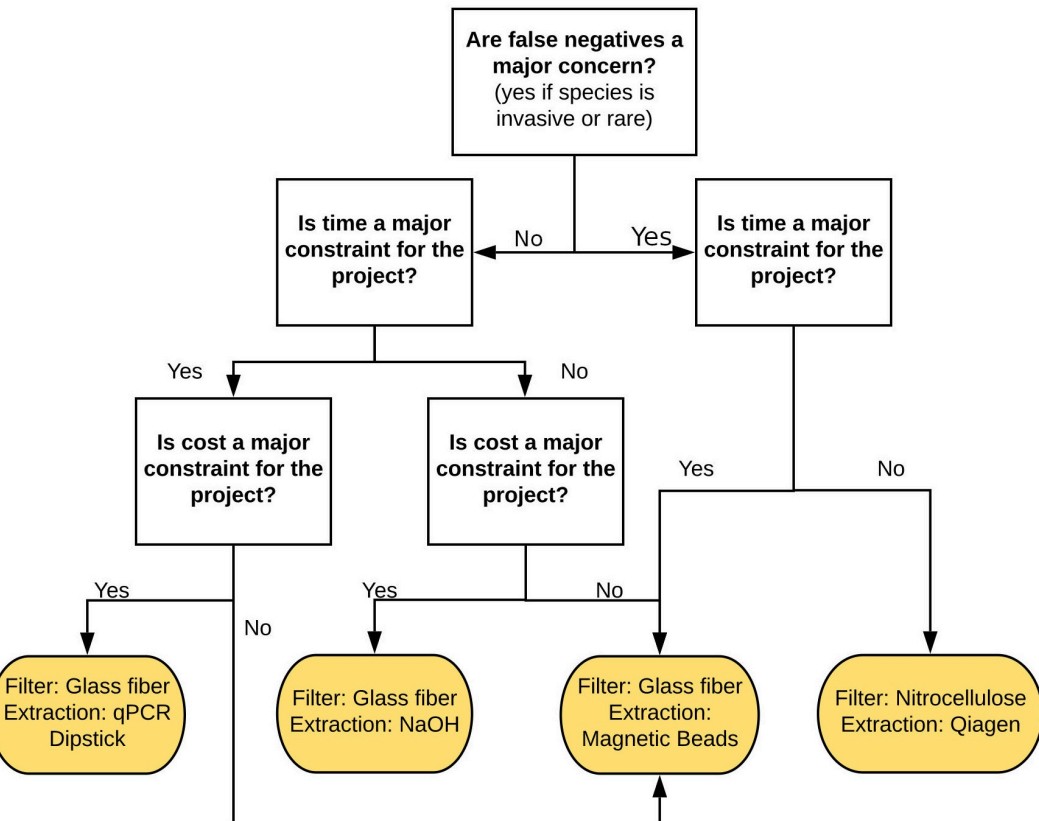

**Fig 8. Decision tree for choosing the protocol which will yield the most information given research constraints.** A glass filter is
recommended in most cases, as long as the focus isn't maximizing DNA yield with no time or cost constraints. Magnetic beads also
are advised in general for its balance between DNA yield and time to process the samples, while cost can be mitigated by producing
magnetic beads solution in-house (~$0.55/mL) instead of buying Ampure XP ($15–$70/mL) [31].

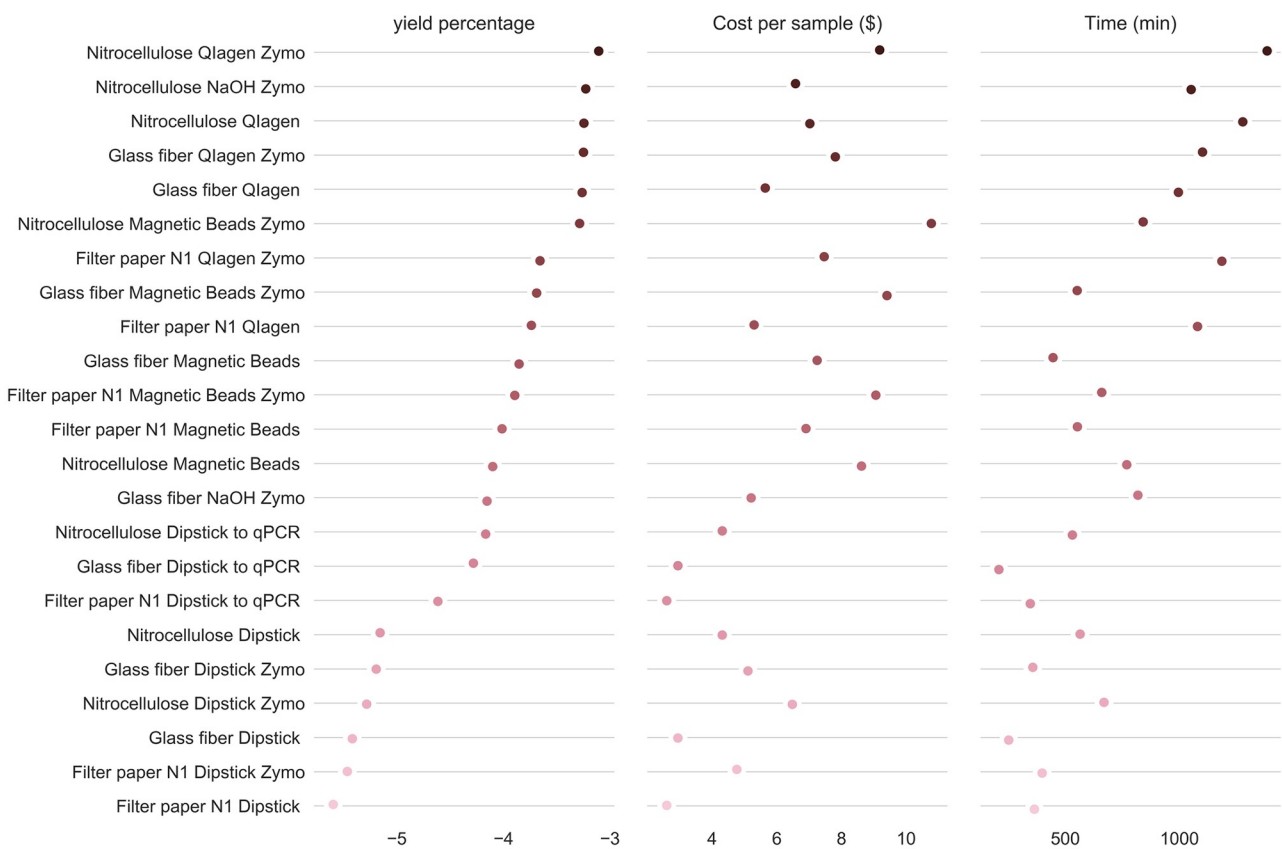

**Fig 9. Comparison between eDNA protocols for DNA yield, cost and time to process 96 samples.** Methods were sorted by yield and shown in $\log_{10}$ scale.

sorting them by DNA yield, which should be the main parameter for the protocol selection. Then, given an eDNA assay sensitivity cutoff, we can choose a protocol that is fast and cost-effective (Fig 9). To choose the optimal eDNA assay protocol, prior information about the target species' occupancy rates in the locations that will be sampled is beneficial. Even though detecting occupancy is also the goal of a species survey, having this prior information will help managers decide how sensitive the eDNA assay must be in order to have high confidence in negative results for a given site [34]. If there is no prior information about the species distribution in the region, we would assume a uniform distribution of the species. Otherwise, if prior information shows a species to be common, faster and less sensitive tests can be used. Meanwhile, if the species is rare, one should avoid false negatives and therefore must use the most sensitive protocol. This is the most crucial question for the project design since the least desirable outcome is for the species to be completely undetected when there is a high false-negative rate. Another undesirable outcome is undersampling the study region. In those cases, even if there is high confidence in the species' presence at a given site, those sites may not be representative of the study region, therefore leading to under or overestimation of the species distribution. Sampling more sites or having higher confidence in a given site is the most complicated choice when designing an eDNA survey. Due to the impact of sampling size on the conclusions of a survey, we consider processing time to be the second main factor when choosing an eDNA assay protocol. Lastly, we consider cost to be the third main factor. In cases where

sensitivity is not a major factor, we can use cheaper assays to largely improve the sampling effort, providing robustness to the findings of the survey.

## Supporting information

**S1 Fig. Correlation between DNA yield and initial eDNA concentration.** Blue dots—median value; vertical lines—95% CI; horizontal line—linear regression between protocol DNA yield and input DNA, with both axes being represented in $\log_{10}$ scale. P(amp)—probability of amplification.
(TIF)

**S2 Fig. Probability of amplification as a function of the input DNA concentration.** Dots—probability of amplification from DNA spiking experiment; line—logistic fit to data points.
(TIF)

## Author Contributions

**Conceptualization:** Thiago M. Sanches.

**Data curation:** Thiago M. Sanches.

**Formal analysis:** Thiago M. Sanches.

**Funding acquisition:** Andrea D. Schreier.

**Investigation:** Thiago M. Sanches, Andrea D. Schreier.

**Methodology:** Thiago M. Sanches.

**Project administration:** Andrea D. Schreier.

**Software:** Thiago M. Sanches.

**Validation:** Thiago M. Sanches.

**Visualization:** Thiago M. Sanches.

**Writing – original draft:** Thiago M. Sanches, Andrea D. Schreier.

**Writing – review & editing:** Thiago M. Sanches, Andrea D. Schreier.

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
