## [Decision Letter · Decision Letter 0]

16 Mar 2020

PONE-D-19-35129

Optimizing an environmental DNA protocol for monitoring threatened Chinook Salmon in the San Francisco Estuary: balancing sensitivity, cost and time.

PLOS ONE

Dear Mr. Sanches,

Thank you for submitting your manuscript to PLOS ONE. After careful consideration, we feel that it has merit but does not fully meet PLOS ONE’s publication criteria as it currently stands. Therefore, we invite you to submit a revised version of the manuscript that addresses the points raised during the review process.

I got the reviewers’ comments for the manuscript. The both reviewers are positive for the publication but found some revision points, e.g., the structure of the manuscript needs to be reorganized, the results were not fully discussed, and sufficient explanations are lacking in some paragraphs. I totally share the comments and recommend to revise for reconsidering the publication.

We would appreciate receiving your revised manuscript by Apr 30 2020 11:59PM. To enhance the reproducibility of your results, we recommend that if applicable you deposit your laboratory protocols in protocols.io, where a protocol can be assigned its own identifier (DOI) such that it can be cited independently in the future. For instructions see: http://journals.plos.org/plosone/s/submission-guidelines#loc-laboratory-protocols

We look forward to receiving your revised manuscript.

Kind regards,

Hideyuki Doi

Academic Editor

PLOS ONE

Journal Requirements:

'We sampled water in accordance with the University of California Davis Institutional Animal Care and Use Committee (USDA registration: 93-R-0433, PHS Animal Assurance A3433-01) under the protocol number #20608.'

a. Please amend your current ethics statement to confirm that your named ethics committee specifically approved this study.

b. Once you have amended this statement in the Methods section of the manuscript, please add the same text to the “Ethics Statement” field of the submission form (via “Edit Submission”).

For additional information about PLOS ONE submissions requirements for ethics oversight of animal work, please refer to http://journals.plos.org/plosone/s/submission-guidelines#loc-animal-research

4. Please ensure that you refer to Figure 9 in your text as, if accepted, production will need this reference to link the reader to the figure.

5. We note you have included a table to which you do not refer in the text of your manuscript. Please ensure that you refer to Table 1 in your text; if accepted, production will need this reference to link the reader to the Table.

Additional Editor Comments (if provided):

I got the reviewers’ comments for the manuscript. The both reviewers are positive for the publication but found some revision points, e.g., the structure of the manuscript needs to be reorganized, the results were not fully discussed, and sufficient explanations are lacking in some paragraphs. I totally share the comments and recommend to revise for reconsidering the publication.

Reviewers' comments:

Reviewer's Responses to Questions

**Comments to the Author**

1. Is the manuscript technically sound, and do the data support the conclusions?

Reviewer #1: Yes

Reviewer #2: Yes

2. Has the statistical analysis been performed appropriately and rigorously? 

Reviewer #1: N/A

Reviewer #2: Yes

3. Have the authors made all data underlying the findings in their manuscript fully available?

Reviewer #1: Yes

Reviewer #2: Yes

4. Is the manuscript presented in an intelligible fashion and written in standard English?

Reviewer #1: Yes

Reviewer #2: Yes

5. Review Comments to the Author

Reviewer #1: The manuscript entitled “Optimizing an environmental DNA protocol for monitoring threatened Chinook Salmon in the San Francisco Estuary: balancing sensitivity, cost and time” examined the effects of eDNA experimental treatments on results and provided a guideline to choose an eDNA protocol. Although many previous studies have deeply and widely investigated the effects of treatments on the results by various approaches, the present study is new in the context of evaluation of the combined effects of each treatment by using models. So, I believe that this study provides useful information to the researchers if published. On the other hand, the structure of the manuscript needs to be reorganized, the results were not fully discussed, and sufficient explanations are lacking in some parts. I made some remarks below and believe that the manuscript will be worthy to be published if the authors fairly addressed the issues I raised.

The structure of the manuscript

The main body of the manuscript should be reorganized.

(1) The methods section should come before the results, as the focus of this study is to evaluate the effects of each step over the eDNA experimental procedure on the results. In other words, the method is pretty important for this study.

(2) Subsectioning in the Introduction is not necessary.

(3) Two subsections (estuary waters and Chinook salmon) can be moved to the materials and methods section if they are just a case/example. Or, if there are any special biological reasons to employ them in this study, how this study contributes to address those biological questions should be discussed in the discussion part.

(4) The results and discussions should be separated. The interpretations of the results of this study are not very very easy, so the readers can understand the results more easily if the authors provide more detailed explanations on each result. Then, the authors can make sufficient discussions by combining all the results.

(5) The current subsection titles in the results and discussion are too long. They should be renamed by making the discussing points clear.

Abstract

Methods and results are not mentioned. They should be included in Abstract.

Introduction

For each of the four steps that the authors determined, they should be explained fairly about what a problem can be, what should be examined and how those problems are addressed.

Results

Filter type (Fig. 1); the authors did not test some types of filters that are also commonly used in eDNA studies such as nylon and polyvinylidene difluoride (PVDF) filters. The authors should explain why they chose paper filter, glass fibre and nitrocellulose.

Filtration time (Fig. 3); the authors did not present the pore sizes of the filters examined. Thus, the relevance of the results could not be evaluated.

DNA extraction; the dipstick direct to qPCR approach seems to be in contradiction with the fact that the authors made account of PCR inhibitor removal. The explanation on inclusion of this approach should be briefly provided.

The starting point of the flow (Fig. 7; does the interest species occupy a high percentage of the sampled sites?) seems a bit special situation in practice. The researchers who employ the eDNA technique generally want/need to estimate the distribution and/or abundance of the species of interest or species community, so it is as if that starting question is the final goals of a study. In addition, the term of a high percentage is unclear (high abundance, density, biomass …?). Guessing from the title of this manuscript, either cost or time may be probably more adequate.

The authors finally provided the flow to make a decision on how to perform the eDNA analyses (Fig. 7). If this study were conducted to monitor the endangered Chinook salmon in the San Francisco Estuary, the authors would be able to show the practical strategy (and preferably the results if they have).

Discussions

As a large number of technical studies for eDNA analyses have been published, there should be many aspects to be discussed, which are definitely lacking in the present manuscript. The authors apparently aimed at developing a kind of universal guideline to make a decision on choosing an eDNA analysis protocol, so the broad discussions including for example comparisons with previous studies are necessary, which also increases the scientific value of this study.

Reviewer #2: Authors aimed to optimize environmental DNA pipeline, filtration, DNA extraction, PCR inhibitor removal and DNA amplification. The results can be useful not only for the eDNA users but also the person who is considering to start eDNA monitoring. Particularly, evalutating specific costs and bifurcation chart are the novelty of this article. However, eDNA studies are updating day by day, we need ongoing updates of this conclusion. And you should explain what kinds of PCR inhibitor were included in your samples and excluded from your samples.

6. PLOS authors have the option to publish the peer review history of their article (what does this mean?). If published, this will include your full peer review and any attached files.

Reviewer #1: No

Reviewer #2: Yes: Hiroki Mizumoto

---

## [Author Response · Author response to Decision Letter 0]

1 May 2020

Academic editor:

File naming was edited to comply with the style requirements. We hopefully have no divergences from the style requirements now.

'We sampled water in accordance with the University of California Davis Institutional Animal Care and Use Committee (USDA registration: 93-R-0433, PHS Animal Assurance A3433-01) under the protocol number #20608.'

a. Please amend your current ethics statement to confirm that your named ethics committee specifically approved this study.

We amended the ethics statements to show that the study was approved by UCD IACUC on line 127-129: 

“Holding juvenile Chinook salmon in captivity to sample water in this study was approved by the University of California Davis Institutional Animal Care and Use Committee (USDA registration: 93-R-0433, PHS Animal Assurance A3433-01) under the protocol number #20608.”

b. Once you have amended this statement in the Methods section of the manuscript, please add the same text to the “Ethics Statement” field of the submission form (via “Edit Submission”).

The ethics statement was also amended in the submission form. 

 The title was also amended in the submission form. 

4. Please ensure that you refer to Figure 9 in your text as, if accepted, production will need this reference to link the reader to the figure

Figure 9 is now referenced on line 455

5. We note you have included a table to which you do not refer in the text of your manuscript. Please ensure that you refer to Table 1 in your text; if accepted, production will need this reference to link the reader to the Table.

Table 1 is now referenced on line 268

Additional Editor Comments (if provided):

I got the reviewers’ comments for the manuscript. The both reviewers are positive for the publication but found some revision points, e.g., the structure of the manuscript needs to be reorganized, the results were not fully discussed, and sufficient explanations are lacking in some paragraphs. I totally share the comments and recommend to revise for reconsidering the publication.

We modified the manuscript to address the points made by the editor and the reviewers. We agreed with the comments in all accounts. We believe that the manuscript is now more readable, more informative, and its conclusions more useful to the public.

Reviewer #1

The structure of the manuscript

The main body of the manuscript should be reorganized.

(1) The methods section should come before the results, as the focus of this study is to evaluate the effects of each step over the eDNA experimental procedure on the results. In other words, the method is pretty important for this study.

The order of the sections was rearranged and now the methods section is after the introduction.

(2) Subsectioning in the Introduction is not necessary.

Sub Sectioning in the introduction was removed and the flow of the reading in the introduction section was ameliorated

(3) Two subsections (estuary waters and Chinook salmon) can be moved to the materials and methods section if they are just a case/example. Or, if there are any special biological reasons to employ them in this study, how this study contributes to address those biological questions should be discussed in the discussion part.

We opted to shorten the Chinook salmon and estuarine waters section and keep that information in the introduction without the subsections. We believe in this case, as eDNA is highly dependent on the tested environment. A brief mention of our system in the introduction would improve readability.

(4) The results and discussions should be separated. The interpretations of the results of this study are not very very easy, so the readers can understand the results more easily if the authors provide more detailed explanations on each result. Then, the authors can make sufficient discussions by combining all the results.

We separated the results from the discussion while improving the explanation of the results. We opted to provide the discussion section without subheading while including comparisons of our results to the literature. 

(5) The current subsection titles in the results and discussion are too long. They should be renamed by making the discussing points clear.

We shortened the subsection titles while also clarifying the goal of each subsection.

Abstract

Methods and results are not mentioned. They should be included in Abstract.

We amended the abstract to include methods and results in a concise manner on lines 22, 27-29

Introduction

For each of the four steps that the authors determined, they should be explained fairly about what a problem can be, what should be examined and how those problems are addressed.

We added information about the specifics of each step, mentioning the problems in each step and how to address them.

Results

Filter type (Fig. 1); the authors did not test some types of filters that are also commonly used in eDNA studies such as nylon and polyvinylidene difluoride (PVDF) filters. The authors should explain why they chose paper filter, glass fibre and nitrocellulose.

The choice of filter materials is now explained on lines 399-404. Tested filter materials were chosen as nitrocellulose has, in general, the highest DNA retention rate in the literature, while glass fiber is currently the most common filter material while Whatman paper was tested as an experimental material which main advantage is the availability and its cost being a fraction of other filter materials. 

Filtration time (Fig. 3); the authors did not present the pore sizes of the filters examined. Thus, the relevance of the results could not be evaluated.

Pore sizes in Figure 3 are now reported and the reasoning for the chosen pore sizes is described on line 399-404

DNA extraction; the dipstick direct to qPCR approach seems to be in contradiction with the fact that the authors made account of PCR inhibitor removal. The explanation on inclusion of this approach should be briefly provided.

An explanation for the observation and reasoning for including this approach is now described on line 242, 250-254.

The starting point of the flow (Fig. 7; does the interest species occupy a high percentage of the sampled sites?) seems a bit special situation in practice. The researchers who employ the eDNA technique generally want/need to estimate the distribution and/or abundance of the species of interest or species community, so it is as if that starting question is the final goals of a study. In addition, the term of a high percentage is unclear (high abundance, density, biomass …?). Guessing from the title of this manuscript, either cost or time may be probably more adequate.

We addressed this question by rewording the figure, changing from species abundance to confidence in negative tests (“are false negatives a major concern”). We also noticed that the figure explanation in the text was subpar and we addressed it by justifying every section of the decision tree. Therefore we expect the figure to be more legible and contain more useful information for our target audience. Changes can be seen on lines 457-472

Discussions

As a large number of technical studies for eDNA analyses have been published, there should be many aspects to be discussed, which are definitely lacking in the present manuscript. The authors apparently aimed at developing a kind of universal guideline to make a decision on choosing an eDNA analysis protocol, so the broad discussions including for example comparisons with previous studies are necessary, which also increases the scientific value of this study.

We added comparative information with other studies and made more clear the novel observations of our study compared to others. Changes are described on lines 407-417

Reviewer #2:

 Authors aimed to optimize environmental DNA pipeline, filtration, DNA extraction, PCR inhibitor removal and DNA amplification. The results can be useful not only for the eDNA users but also the person who is considering to start eDNA monitoring. Particularly, evaluating specific costs and bifurcation chart are the novelty of this article. However, eDNA studies are updating day by day, we need ongoing updates of this conclusion. And you should explain what kinds of PCR inhibitor were included in your samples and excluded from your samples.

We added the types of PCR inhibitors we would expect in our samples on line 67 and lines 97-98. We added a comment about the fast evolution of the field and how our data analysis and study design might still be useful even if our methods get obsolete lines 411-413.

---

## [Editor Report · Decision Letter 1]

7 May 2020

Optimizing an eDNA protocol for estuarine environments: balancing sensitivity, cost and time

PONE-D-19-35129R1

Dear Dr. Sanches,

We are pleased to inform you that your manuscript has been judged scientifically suitable for publication and will be formally accepted for publication once it complies with all outstanding technical requirements.

With kind regards,

Hideyuki Doi

Academic Editor

PLOS ONE

Additional Editor Comments (optional):

I carefully checked the revised manuscript as well as the response letter. I agree the revisions according to the reviewers’ comments and now can recommend to publish the paper in PLOS ONE.
---

## [Editor Report · Acceptance letter]

12 May 2020

PONE-D-19-35129R1 

Optimizing an eDNA protocol for estuarine environments: balancing sensitivity, cost and time 

Dear Dr. Sanches:

I am pleased to inform you that your manuscript has been deemed suitable for publication in PLOS ONE. Congratulations! Your manuscript is now with our production department. 

With kind regards,

on behalf of

Dr. Hideyuki Doi 

Academic Editor

PLOS ONE